# Colorimetric Sensor Array for Monitoring, Modelling and Comparing Spoilage Processes of Different Meat and Fish Foods

**DOI:** 10.3390/foods9050684

**Published:** 2020-05-25

**Authors:** Lisa Rita Magnaghi, Federica Capone, Camilla Zanoni, Giancarla Alberti, Paolo Quadrelli, Raffaela Biesuz

**Affiliations:** 1Department of Chemistry, University of Pavia, Via Taramelli 12, 27100 Pavia, Italy; lisarita.magnaghi01@universitadipavia.it (L.R.M.); federica.capone01@universitadipavia.it (F.C.); camilla.zanoni01@universitadipavia.it (C.Z.); giancarla.alberti@unipv.it (G.A.); paolo.quadrelli@unipv.it (P.Q.); 2Unità di Ricerca di Pavia, INSTM, Via G. Giusti 9, 50121 Firenze, Italy

**Keywords:** sensor array, food safety/food waste, naked-eye detection, protein food spoilage, multi-purpose device, three-way PCA

## Abstract

Meat spoilage is a very complex combination of processes related to bacterial activities. Numerous efforts are underway to develop automated techniques for monitoring this process. We selected a panel of pH indicators and a colourimetric dye, selective for thiols. Embedding these dyes into an anion exchange cellulose sheets, i.e., the commercial paper sheet known as “Colour Catcher^®^” commonly used in the washing machine to prevent colour run problems, we obtained an array made of six coloured spots (here named Dye name-CC@). The array, placed over the tray containing a sample of meat or fish (not enriched at any extend with spoilage products), progressively shows a colour change in the six spots. Photos of the array were acquired as a function of time, RGB indices were used to follow the spoilage, Principal Component Analysis to model the data set. We demonstrate that the array allows for the monitoring the overall spoilage process of chicken, beef, pork and fish, obtaining different models that mimic the degradation pathway. The spoilage processes for each kind of food, followed by the array colour evolution, were eventually compared using three-way PCA, which clearly shows same degradation pattern of protein foods, altered only according to the different substrates.

## 1. Introduction

Since the first pioneering work by Dainty [1], the control of food quality has risen in its importance and nowadays it is of great concern in our daily life because of the ever-increasing demand for good quality and hygienic food products [2,3,4]. Consumers’ requests for mildly preserved, minimally processed, easily prepared and ready-to-eat “fresher” foods, together with the globalization of the food business, and the logistics of distribution from processing centres [5,6,7] make food quality control a significant challenge. In this scenario, the traditional methods for determining food freshness, based on physical, chemical, microbiological measurement and human sensory evaluation [8,9], have proven insufficient and need to be updated or even replaced by rapid, low cost and non-destructive techniques.

Since this theme has been on the front burner in the last years, a large variety of devices have been proposed so far to monitor food freshness to comply with market demand. We can include in this category biosensors, electronic devices (e-nose and e-tongue) [9] and colourimetric sensors [3,5,10,11].

Colourimetric sensors turn out to be highly attractive since they are capable of changing colour thanks to a reaction with volatile compounds produced in the headspace of packaged meats. Either included directly on packaging or attached with an on-package sticker, they offer the simplest, most practical, instrument-less way for monitoring freshness, directly by the naked eye [12].

The recent literature presents many examples that follow this idea, for instance, immobilized bromocresol green as a fish spoilage indicator [3], or a mixed pH dye-based indicator as a “chemical barcode” [5,10], all based on pH indicators with different *logK*_a_ values.

Nevertheless, in the literature on this topic, even in the most recent papers, some common weak points can be individuated. Firstly, as already assumed in previous work [12], the attention is always focused on the detection and identification of biogenic amines (BA) in the headspace [13,14,15]. In our opinion, this approach is not correct, because other byproducts are produced, even in large amounts, during spoilage and have to be considered. In particular, in the early stages of spoilage, when meat is a safe product, the chemical spoilage index (CSI) is associated with the consumption of glucose and lactic acid, and production of EtOH, 3-methyl-1-butanol and free fatty acid, particularly acetic acid, which, together, make up the dominant Volatile Organic Compound (VOC) [16,17]. Moreover, both meat and fish pH is buffered (as it is in all biological matrices) at a value around neutrality and, under these conditions, BAs are in their protonated form so they cannot fly and be detected in the headspace [12].

The need for algorithms or electronic devices [8,11,18] and, consequently, the impossibility of naked eye analysis represents another critical weakness of many of the proposed devices. We do believe that these kinds of sensors must be tested under conditions as close as possible to real conditions. Consequently, we choose to work not on enriched or simulated samples [14,19], but in a selling tray containing a typical amount of food, stored in an aerobic atmosphere, sealed with conventional food quality films and in a reasonable range of time [20]. Almost none of the proposed devices, to our knowledge, have been simultaneously and extensively tested on different meats and fish.

With this in our mind, we developed a colourimetric array [12] based on five pH indicators with *logK*_a_ values around neutrality [21,22,23] and a specific dye for thiols, an abundant class of volatile byproducts never considered in freshness monitoring devices, to our knowledge. The crucial points in the development of our array were rapidity, efficiency, low price, naked eye analysis from untrained people and testing in real conditions. Finally, to demonstrate the applicability of our device to a large variety of proteins, chicken, beef, pork meat and cod fillets were tested in replicate at ambient temperature.

The final idea is to develop an intelligent label that gives an individual easy to interpret and reliable information, well beyond the expiry date that is adaptable to different foods with minimal changes. The industrial application represents the main scenario for such a device, due to the robust reliability and the very reduced costs of label implementation; for this reason, polymeric support is currently under study [12].

## 2. Materials and Methods

### 2.1. Materials and Chemicals

All reagents were Analytical Reagents grade. *m*-cresol purple (1), *o*-cresol red (2), bromothymol blue (3), thymol blue (4), chlorophenol red (5), Ellman’s reagent (6) were purchased from Carlo Erba or Sigma Aldrich.

Dylon Colour Catcher^®^ was bought in a local supermarket.

Beast poultry meat in slices, sliced beef, pork and cod fillets were bought in a local supermarket (UNES Supermarkets, via Fratelli Cervi, 11 27100 Pavia, Italy), the same day of the delivery from the central slaughterhouse or fish market, a few moments after the food was put on the shelf.

A Smartphone Samsung Galaxy S7 was used to acquire pictures of the array; a portable led light box was used to guarantee the reproducibility of the photos. We also used a Puluz Photography Lightbox, Shenzhen Puluz Technology Limited, already shown elsewhere [12].

In this study we employed the sensor array already proposed in a previous work for chicken meat degradation monitoring [12]; the sensitive part, the solid support and the experimental setup were kept unchanged, but we extended its applicability towards different food samples.

### 2.2. The Sensitive Part of the Sensors

The first five sensitive molecules selected for the array are acid-base indicators: m-cresol purple (1) (log*K*_a1_ = 8.32, log*K*_a2_ = 1.57), o-cresol red (2) (log*K*_a1_ = 8.20, log*K*_a2_ = 1.11), bromothymol blue (3) (log*K*_a_ = 7.1), thymol blue (4) (log*K*_a1_ = 8.90 log*K*_a2_ = 1.50), chlorophenol red (5) (log*K*_a_ = 6.0), and their log *K*_a_ values, as found in [21,22] are reported within parentheses. The log *K*_a_ values are in agreement with those also reported in [23]. The sixth is the 5,5’-dithiobis(2-nitrodibenzoic acid) (6) (DTNB), generally called Ellman’s reagent: it is a molecule with two electron-deficient phenyl groups linked by a sulphydryl bridge. In the presence of thiols, it readily undergoes a trans-sulforation reaction with the reduction of the sulfhydryl group that releases a highly chromogenic product, the 5-thio-2-nitrobenzoate (TNB), with an intense absorption band at 412 nm [24]. All these molecules present one or two, in the case of Ellman’s reagent, permanent negative charges. For convenience, in the following, the order of sensors in the array will keep the same number from one to six as reported above.

### 2.3. The Solid Support of the Array

As a solid support, we choose the Colour Catcher^®^ (CC) a product from the washing market, distributed in Italy by Grey, in England by Dylon, partners of Henkel Company, purchasable in any supermarket.

CC and similar products have become successful for their ability to prevent colour run during the washing operation. For this purpose, one sheet must simply be added into the laundry before starting the desired washing machine cycle. If cloths release colours, they are immediately sorbed by the CC, and not by other fabrics. To exert its properties, the CC must exhibit sequestration properties towards molecules and ions when released by clothes, even in the presence of surfactants and fabric softeners. Since tissue dyes are often anionic, we tested it as an anion exchange device.

The chemicalphysical characterization of the solid phase was already discussed in two previous papers, where Alizarine RedS [25] and the Ellman’s reagent [26] were employed as the sensitive part of two different sensors.

### 2.4. Preparation of the Dye Name-CC@ and Experimental Set-Up

The preparation of the Dye name-CC@ followed the same procedure presented in previous work [12]. The CC was cut in circles of 0.4 cm diameter of approximatively 0.0015 g, obtained with a hole punch for paper. The acid-base form of the dye is important since the ion exchange sorption reaction on the CC causes the dyes to change into their basic colour. Table 1 reports the optimized conditions to prepare the final sensors.

An amount of 1 mL of the dye solution, at the concentration reported in the second column of Table 1, was placed in an Eppendorf tube. After adding the proper amount of acid (reported in the third column), the samples were left to equilibrate overnight at ambient temperature, on a stirring plate. Subsequently, the CC spots were dried and kept ready for use. When needed, they were placed on a stripe of Scotch 3M Magic Tape and put over the tray to offer the free side to the inner part of the packaging. Except for Scotch 3M Magic Tape*,* other adhesives like glue, or normal tape were not suitable, since they all release acidic substances that change the colour of the dyes into their acidic colours.

Food samples were purchased, as already stated, in a local supermarket. We bought directly ready trays made of a plastic container (PP) for food and covered with low-permeability polyethylene plastic film. The trays were taken from the shelf just a few moments after preparation (based on previous discussions with the head of the butcher’s and fishmonger’s departments) to ensure the homogeneous lifetime of all samples. Within ten minutes, the samples were in the lab, the plastic film was removed, the stripes with sensors were placed over the tray and a new plastic film was fixed around the tray. The samples were placed under the hood, depending on the type of experiment. Figure 1 shows a picture of the experimental setup for different food samples.

### 2.5. Colour Analysis

The photos were acquired by a Smartphone Samsung Galaxy S7 in a lightbox to ensure a constant and reproducible light exposition. The reproducibility refers to differences among photos on the same spot and was much better with this method than using the flash mode of the camera. In our case, it is important to evaluate one colour that turns into another, and the RGB triplet gives much more information than other indices or colour spaces.

Despite questions related to sensitivity, the final shape and dimension of each sensor ensure a homogenous final shade that makes it possible to acquire the colour features without any filter or any other strategy of sampling. The GIMP software (open source program https://www.gimp.org/) was employed, which allows for the definition of the area of the photo to be analyzed, usually over the entire spot, and gives the average values of RGB triplets for each sample.

### 2.6. Real Sample Analysis

For each food, five samples, with similar weights (reported in Table 2) were acquired, prepared, following the procedure in Section 2.3, and kept at ambient temperature.

The arrays placed in the headspace were shot at different times during the degradation. Table 3 reports the timelines of the analysis for each food. As can be seen, the duration of the analysis and the frequency of picture acquisition depended mainly on the type of food, as shown in Table 3. The choice of these timelines for the different foods was based, on the one hand, on the wide variation in the perishability of the different substrates and, on the other hand, on the awareness that it makes no sense to test a very advanced degradation state. Consider that, in any case, the final time in Table 3 always corresponds, for each food, to an advanced degradation step. There is no need to test spoilage after many days or weeks. What we are looking for is a sensor able to reveal when protein degradation begins, not to individuate full-blown spoilage. 

For each sensor in each photo, the corresponding RGB triplets were acquired. The Principal Component Analysis (PCA) was performed by centring, but not scaling, the data since the RGB triplets are intrinsically scaled (RGB values vary from 0 to 255) to model the spoilage. The open-source Chemometric Agile Tool (CAT) program was employed both for PCA and later for three-way PCA analysis. (http://www.gruppochemiometria.it/index.php/software/19-download-the-r-based-chemometric-software). Other chemometric techniques could be used, such as classification methods, but this is not our purpose, so we used PCA only to visualize and analyze the colour evolution of the sensor array.

Moreover, for each food, the most informative receptors, which are the ones with the highest contributions in the loading plot, were individuated. This preliminary investigation can give some suggestions about which receptor should be selected or discarded in a large-scale implementable device based on two or three sensing units. Other tools, such as classification and/or modelling, will be adopted with a massive number of samples, which are mandatory in those cases [27], at which point the choice of sensors will be performed and the final support (not CC) will be developed.

### 2.7. Samples for Preliminary Validation

In order to perform a preliminary and “naive” validation of the PCA model, new samples of each food were acquired and prepared with the same procedure, described in Section 2.3. Table 4 reports the mass of the samples used for this validation.

At a given time, the array was photographed and the RGB indices were acquired and projected as an external dataset in the PCA model, in order to verify if the colour evolution is the same and if the location of the new sample was in good agreement with the training set.

### 2.8. Samples for Sensitivity Test

From everyday life experience, we noticed that the amount of meat in the selling tray could widely vary, depending on the type of meat, the quality and even the supermarket. At first, to develop the degradation model, we used samples containing similar amounts of meat, around 300 g for chicken meat and 150 g for beef and pork meat (see Table 2), since these amounts of meat per tray are the most common on the supermarket shelf. For this kind of packaging, the sealed volume below the plastic film is of about 500 cm^3^. When decreasing the amount of sample on the same tray, the analytes dilute progressively and it is relevant to evaluate the sensitivity of the array and to estimate the lowest amount of meat that produces an equal colour evolution.

For this reason, decreasing fractions of meat mass used for the model development, reported in Table 5, were sealed in the common tray and analyzed following the procedure explained before.

### 2.9. Comparison between Different Spoilage Processes

Eventually, we compared the colour evolution observed during the spoilage process of the investigated foods. This investigation has a dual purpose: on the one hand, we want to demonstrate that different protein foods undergo a typical spoilage process, characterized by the same steps, in terms of nutrients attacked and byproducts released; what differs among them is the duration of the steps. This should be understood as further evidence of the reliability and broad applicability of the here-proposed device.

This comparison was performed using three-way PCA, which is the foremost technique to deal with three-dimensional data and, in particular, with the analysis of significantly different samples over time [28]. We used the RGB indices of the sensors in the array (variables), centred but not scaled, to study the degradation, taking into account the composition of different foods (objects) over time (conditions).

For this comparison, we have to use a few tricks: firstly, we have to use the entire array because the selection of the most informative receptors is dependent on the food under investigation, which means that each food has a group of two/three more useful dyes that are different from one another. Secondly, for the difference in perishability and spoilage duration (see Table 3), the shortest timeline must be used. We were forced to refer to the codfish (which must be thrown away after 48 h) to select the acquisition times common to all. The resulting time samples are reported in Table 6.

## 3. Results and Discussion

### 3.1. Evolution of the Colours over Different Foods at Room Temperature

The first step of the monitoring consists of a naked-eye analysis of the colour evolution, shown by the sensor array during the degradation process. We adopted the same procedure for the evolution of chicken meat at different temperatures [8]. As reported in the Materials and Methods section, we registered the array colour evolution of five samples for each food. Figure 2 shows an example of the sensors’ colour evolution for each food.

At first glance, by a simple naked-eye analysis of the colours, we can notice that similar behaviour is detected in all the food, despite the different timelines, because, as already highlighted, the same spoilage process is occurring. In particular, two clear steps can be identified. Firstly, the evolution of alcohol and acids arises from sugar catabolism by bacteria [8], detected by pH indicators that turn their colour from the basic form into the acidic one. For this purpose, four dyes (from one to four) with log*K*_a_ higher than seven have been included in the array. Moreover, we can see that the receptor with the highest log*K*_a_, i.e., Thymol Blue (dye no. 4) [16], is the first one that reacts with acids and completely changes to the yellow acidic form, regardless of the food. Then, dye no. 1 and dye no.2 turn partially to the acidic colour, following their slightly lower log*K*_a_, showing an intermediate shade, depending on the sample. The slowest reaction corresponds to dye no. 3, which has the lowest *logK*_a_ and turns from blue to green, never reaching a complete conversion into the yellow acidic form.

When sugars and their derivates are eventually depleted, the bacteria attack the proteins, releasing amines and thiols. Both these classes of molecules are definitively unwanted and dangerous. However, they differ widely in their volatility, especially in the buffered pH typical of the biological matrix: in fact, amines are present in their protonated form, and can pass into the headspace only at an insignificant amount, as already demonstrated before [8]; on the other hand, thiols are neutral and volatile molecules that are present in great abundance in the headspace. For this reason, during the second step, only a slight increase in the pH is expected, while thiols can be easily detected. Indicator 5 (log*K*_a_ = 6.0), exposed in the acidic yellow form, shows an appreciable colour change to violet (basic form) in all the samples, at different times. However, the other receptors with higher log*K*_a_ remain unchanged. The Ellman’s reagent (6) has been included in the array to detect not only the acid base changes in the meat samples but also the presence of thiols. This receptor reacts in all the foods tested but with very different timings: in chicken and fish samples, an intense yellow colour is observed after 21 h, while in beef and pork meat, the detection occurs only after 48 h.

Finally, by simple naked eye analysis of the colours, many conclusions can be drawn, all in good agreement with previous knowledge about the chemistry of spoilage and the differences in perishability of the investigated foods. To rationalize these preliminary findings and to model the entire spoilage process, PCA was performed on all the RGB data of the samples, as described in Section 2.5.

### 3.2. PCA Models Development and Validation

As explained before, PCA is employed in this case to visualize and model the degradation process of different meats without claims of classification. Moreover, only RGB triplets of the six sensors were used as the training set while the time passed is used only as a label to identify the samples. After model development, two independent samples were used as a test set and projected in the score plot to verify the correct allocation and, thus, preliminarily validate the model.

Figure 3 shows the score plots on the first two components for all the foods, reporting both the training set (coloured spots) and the test set (light blue squares). The loading plots, instead, are reported below in Figure 4.

For meat samples (see Figure 3a–c), three groups can be detected, called SAFE, WARNING and HAZARD. In the case of chicken meat, (see Figure 4a) the clusters are well separated while, for the last two foods (see Figure 4b,c, for beef and pork meats, respectively), the WARNING and HAZARD zones are partially overlapped. We have proved that this separation, in the case of poultry meat, was validated by chemical analyses, both of the headspace and the meat [12]. These findings demonstrated how the attribution of the stages follows the development of acids produced in the early stage, and after the production of thiols, but never of BA, which are only present inside the meat. In all cases, samples in the first hours are located in the lower left part of the score graph; then, during the spoilage process, the samples move to the right side. As a consequence, we can assess that, here, the first component explains the spoilage, but PCA2 is still the fundamental component for the separation of the clusters. During the WARNING step, the score of PC2, generally increases, while it decidedly decreases for samples that fall into the HAZARD zone.

As for model validation, we can observe that all the samples in the test set are located in the right cluster, near training samples with similar times. This evidence is essential both to assure the reproducibility of the sensors’ responses and to validate the models developed by PCA.

What is also very clear is that the score plot of fish (see Figure 3d) is utterly different compared to other meat, a diversity much more evident than the spot evolution of Figure 2. It is common knowledge that fish is an extremely perishable food and its spoilage process is very fast, even if stored in appropriate conditions. The PCA model on the RGB indices makes it manifest: only two clusters are identified, corresponding to SAFE and HAZARD zone, without any intermediate step. In particular, the SAFE cluster is in the lower part of the graph; as the spoilage goes on, the samples move to the higher part of the graph, i.e., at the high value of PC2, and definitively give a significant contribution to the cluster separation.

After the development and validation of PCA models, a preliminary selection of the most informative dye for each food can be performed, analyzing the colour evolution and the loading plots, as reported in Figure 4.

The most informative receptors are the ones which give the highest contribution in the loading plots, marked as the coloured blocks in the graphs above. It is interesting to note that some dyes are significant for each food, such as Bromothymol Blue (3) and Ellman’s reagent (6); conversely, other pH indicators can be informative or not, depending on the type of food. This selection has to be considered as a preliminary step towards a large-scale implementable device with a minimum number of sensors. It should be confirmed by analyzing a new set of samples with only these receptors and developing and validating these new reduced PCA models. These experiments are out of our reach at this point, mainly because the solid support here presented is neither final nor suitable for industrial application, and a new material is currently under investigation [20].

### 3.3. Sensitivity

The goal of these experiments was to estimate the minimum mass of food characterized by an evident and similar colour evolution as that previously observed in training and test samples. Such an estimation cannot be performed by a simple naked-eye analysis of the colour evolutions with sufficient certainty, but PCA can be useful, at least for a preliminary estimation. For this reason, for each food, we ran PCA on the RGB indices of sensors exposed over some subsamples of reduced mass, as reported in Section 2.7 (See Table 5), and we analyzed the resulting score plots, which are reported in Figure 5.

The trays with similar mass to training and test samples, i.e., 100%, were considered the reference tray and their degradation process was used as the “reference pathway” in the score plot. We noticed that, down to a certain amount of mass, colour evolution overlaps that of the reference. It means that samples in the score plot fall near the reference ones, following the same “pathway”. Below this mass value, the colour evolution significantly differs, resulting in a different allocation in the score plot. In Figure 6, for each food, only reference points and samples above critical mass are highlighted and labelled with the corresponding reference mass percentage; the critical mass percentage corresponds to 50% in the case of chicken and pork, and 25% for beef. As previously argued, this is not an absolute value. Indeed, the analyte concentration depends on the amount of food and headspace volume (typically around 500 cm^3^). The sensitivity of the array, in terms of meat mass, can be sharply improved by using smaller trays with lower volume.

### 3.4. Comparison between Different Spoilage Process by Three-Way PCA

We performed a comparison between the spoilage processes of the different protein foods, as described in Section 2.8, using three-way PCA. Five samples for each food under investigation were employed as objects (20 in all), the acquisition times, reported in Table 6, were identified as conditions (six in all), while the RGB triplets of the entire array were used as variables (18 in all).

The variance percentage explained after unfolding, reported in Table 7, is definitively higher than the value related to the Tucker model (50.81%). This means that part of the information contained in the dataset is lost when the common degradation process is taken into account. In our opinion, this loss is mainly due to the significant difference in spoilage duration for the considered types of meat. In fact, as already observed, after 48 h, cod fillets were in an advanced state of decomposition, while spoilage was still ongoing for the other samples. This information is inevitably lost by analyzing the common degradation. Nevertheless, the percentage of explained variance is fully satisfying, considering the high variability of the system, the types of data employed for the analysis, and the necessary reduction in the data set.

Typically, the results of three-way PCA are displayed either in three different loading plots, the plot of objects, conditions and variables or in the triplot, which shows all the loading values altogether. In this case, the interpretation of results is much easier when analyzing the separated loading plots, presented in Figure 6, rather than the triplot.

In the plot of objects (Figure 6a), different protein foods are separated along the *x*-axis (Axis 1). In particular, fish samples occupy the right part of the graph; chicken samples are in the centre, while beef and pork are on the left. These last two have a slightly different loading values on the *y*-axis (Axis 2), but they are too near to be distinguished with sufficient certainty. Comparing this graph with the plot of variables (Figure 6c) we can conclude that the most informative receptors in object separation are the ones with a higher loading value on Axis 1, in the foreground, and therefore are *o*-cresol Red (2), Bromothymol Blue (3) and Ellman’s reagent (6). This conclusion is in good agreement with what was previously observed during the analysis of each degradation (Section 3.1 and Section 3.2): pH indicators, namely (2) and (3), do not turn completely to an acidic colour. However, they show a different behavior according to the food under investigation. At the same time, Ellman’s reagent reacts at different times in different types of meat or fish.

In the plot of conditions (Figure 6b), times are well separated along the *y*-axis: the very first degradation steps are located in the lower part of the graph, while, during the spoilage process, the loading values on Axis 2 increase. Again, from the plot of variables (Figure 6c), the most critical receptors in conditional separation are identified, based on the value of the loading on Axis 2, in the background (*m*-cresol Purple (1), Bromothymol Blue (3) and Thymol Blue (4)). We compared the spoilage profile of protein foods with widely different compositions and perishability. Even if the number of conditions is reduced, even if we are well aware that these results are only an indication, they are profoundly coherent and have a precise sense. First, the order of perishability is maintained as fish, chicken, beef and pork (see Figure 4a) secondly, the conditions, even if compressed, clearly show a clear difference between the early stage and the subsequent stages.

### 3.5. Future Developments

This work represents a sort of completion and integration of the first presented array [12], initially applied only to chicken meat. The original idea is anyhow covered by a patent [29].With the present work, we demonstrated not only that our monitoring strategy can be applied to a wide variety of protein foods but, even more, that the array can be adapted and optimized for the food of interest by selecting the proper receptors.

The promising results obtained using the here-presented array encourage us to develop a prototype of an intelligent label, suitable for industrial production and large-scale application. The original idea and the future implementation of the array are presently under patent [29,30]: here, the dyes are not fixed on the CC but are covalently bound to a polymeric matrix, giving rise to a more suitable material to be implemented into a label for industrial application.

In the development of the new prototype, another key point is the reduction in the receptors. As we demonstrated in this work, the selection of an adequate number of dyes must relate to the food of interest. The final number of receptors could be one, two or three at most, depending on the chemistry of spoilage and the perishability of the sample.

Only in this further stage will complete modelling and classification with an adequate number of samples and independent validation analyses be performed [27], and these processes are already planned.

Eventually, in the final label, the reference colours for each receptor at different degradation steps must be printed next to the sensors to allow consumers to identify the freshness of food samples, with no need for specialized people, instrumental analysis or electronic device support.

## 4. Conclusions

The work presents a colourimetric array to follow the entire spoilage process of chicken, beef, pork meat and cod fillets.

The array is prepared, optimizing the dimension of the spot, the amount of dye and its acid or base form to ensure the best colour development over the meat samples.

The analysis of real samples (not enriched ones) highlighted the performance of the arrays. We followed the evolution of the array colours on different food samples kept in the fridge for a reasonable amount of time and based on our consumers’ experience.

With preliminary experiments, we identified the most informative receptors for each meat and fish.

The RGB triplets of the coloured spots and PCA analysis allow us to monitor the spoilage process and to identify different clusters for each food.

Three-way PCA was used as a proof-of-concept to demonstrate the applicability of the device to different food degradation stages and the existence of a typical degradation process, despite the differences in food composition.

The goal of this work is to demonstrate and highlight that a generalized colourimetric array, based on commercially available receptors and support, can be applied to various food degradation monitoring methods with encouraging results. Moreover, by PCA analysis, degradation processes can be studied more in detail and models can be developed. Eventually, by three-way PCA, the degradation processes of four different types of meat and fish can be summarized in one generalized model.

One idea for the possible implementation and development of this device is presented for an in-field application. Furthermore, the evolution of the array, obtained by fixing the dyes on a solid support, is under study.

## Figures and Tables

**Figure 1 foods-09-00684-f001:**
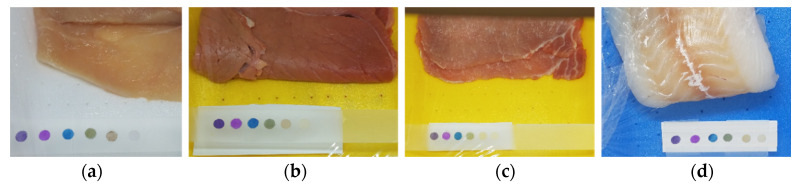
An example of the array placed over trays containing chicken meat (**a**), beef meat (**b**), pork meat (**c**) and codfish (**d**).

**Figure 2 foods-09-00684-f002:**
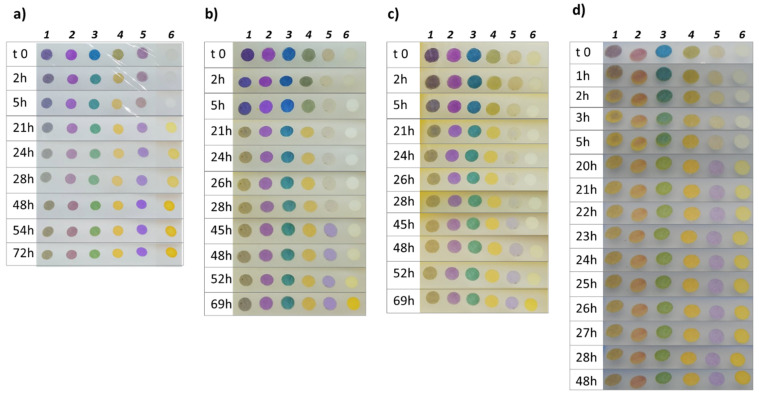
The colour evolution of the sensor arrays over chicken meat (**a**), beef meat (**b**), pork meat (**c**) and cod fillets (**d**) kept at room temperature.

**Figure 3 foods-09-00684-f003:**
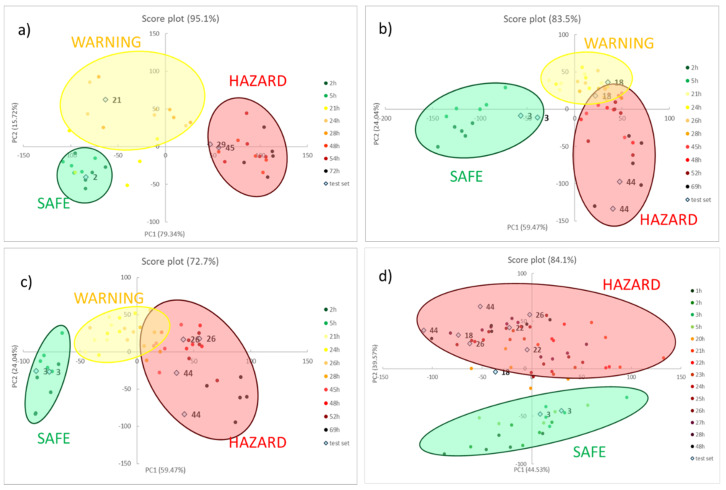
The score plots of the PCA models on the first two principal components, considering all the samples of chicken meat (**a**), beef meat (**b**), pork meat (**c**) and cod fillets (**d**) kept at room temperature. The ellipsoids are exclusively added as a simplification of the different groups: SAFE, WARNING and HAZARD.

**Figure 4 foods-09-00684-f004:**
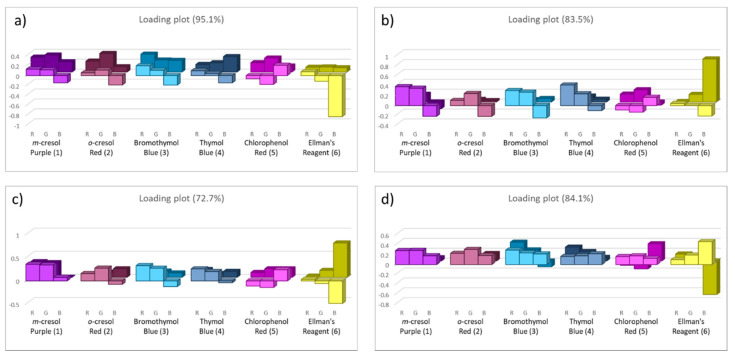
The loading plots of the PCA models on the first two principal components, considering all the samples of chicken meat (**a**), beef meat (**b**), pork meat (**c**) and cod fillets (**d**) kept at room temperature.

**Figure 5 foods-09-00684-f005:**
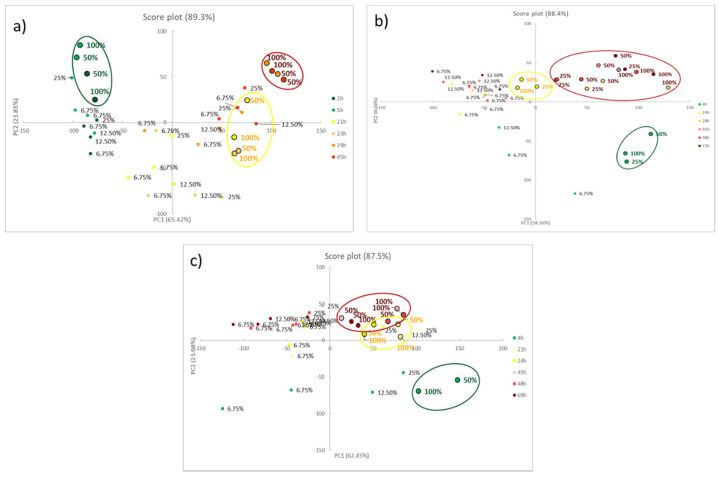
The score plot of the PCA models on the first two principal components, considering samples for sensibility tests of chicken meat (**a**), beef meat (**b**) and pork meat (**c**).

**Figure 6 foods-09-00684-f006:**
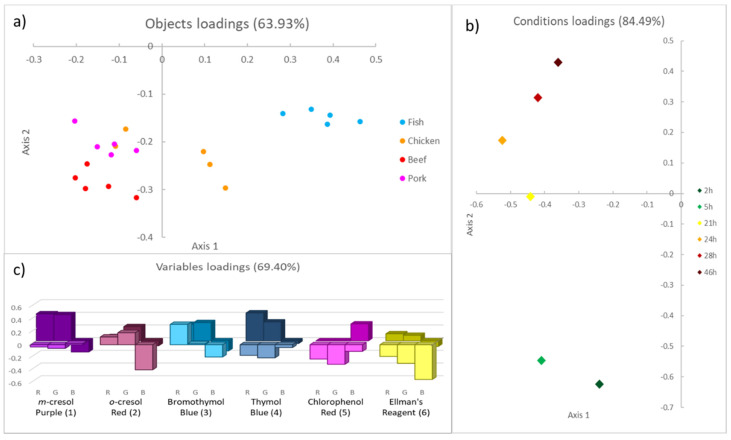
The loading plots of the three-way PCA model on the first two axes: objects loading plots (**a**), conditions loading plots (**b**) and variables loading plots (**c**).

**Table 1 foods-09-00684-t001:** Experimental conditions for the Colour Catcher^®^ (CC) spot sensors preparation.

	Dye Concentration (M)	µL HCl 10^−3^ M
*m*-cresol purple	7 × 10^−6^	20
*o*-cresol red	4 × 10^−6^	40
bromothymol blue	9 × 10^−6^	40
thymol blue	8 × 10^−6^	10
chlorophenol red	7 × 10^−6^	500
Ellman’s reagent	2.4 × 10^−5^	100

**Table 2 foods-09-00684-t002:** Mass of samples of different foods, employed as the training set for PCA model development.

Sample	Chicken Meat (g)	Beef Meat (g)	Pork Meat (g)	Codfish (g)
1	348	148	132	204
2	344	152	134	206
3	342	160	137	200
4	306	161	144	196
5	324	162	141	192

**Table 3 foods-09-00684-t003:** Photo acquisition times for different foods, expressed in hours from the preparation.

Photo	Chicken Meat (h)	Beef Meat (h)	Pork Meat (h)	Codfish (h)
1	2	2	2	1
2	5	5	5	2
3	21	21	21	3
4	24	24	24	5
5	28	26	26	20
6	48	28	28	21
7	54	45	45	22
8	72	48	48	23
9		52	52	24
10		69	69	25
11				26
12				27
13				28
14				48

**Table 4 foods-09-00684-t004:** Mass of samples of different foods, employed as a test set for PCA models.

Sample	Chicken Meat (g)	Beef Meat (g)	Pork Meat (g)	Codfish (g)
1	338	157	138	210
2		162	143	220

**Table 5 foods-09-00684-t005:** Samples of decreasing fractions of meat for sensibility test.

Sample	% Reference Mass	Chicken Meat (g)	Beef Meat (g)	Pork Meat (g)
1	100	300	150	150
2	50	150	75	75
3	25	75	37.5	37.5
4	12.5	37.5	18.75	18.75
5	6.75	18.75	9.375	9.375

**Table 6 foods-09-00684-t006:** Photo acquisition times of samples employed for three-way PCA.

Photo	Three-Way PCA Samples
1	2
2	5
3	21
4	24
5	28
6	48

**Table 7 foods-09-00684-t007:** Cumulative % variance explained after unfolding.

Mode	Axis 1	Axis 1 & 2
Objects	37.92%	63.93%
Variables	42.67%	69.40%
Conditions	53.56%	84.49%

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
