# Peer review of "Colorimetric Sensor Array for Monitoring, Modelling and Comparing Spoilage Processes of Different Meat and Fish Foods"

_foods, 2020, doi:10.3390/foods9050684_

Round 1
Reviewer 1 Report
This manuscript is, indeed, quite interesting. And I can see the huge applicability of a technically simple methodology. I like the scientific idea. Nevertheless, the manuscript, as it is, looks like a mere laboratory session for students. And do not get me wrong. I will clarify this statement. The authors need to create a solid methodology and validate it in order to create a successful methodology that can be used outside the laboratory environment. Only in that case this manuscript will make a real difference. For instance:
- Reproducibility of Dylon Colour Catcher formulation: It is too vague to use a commercial product and not validate its reproducibility. Either manufacture one by yourself following a specific formulation or verify the reproducibility of that commercial product by measuring different lots, different days the same lot, etc.
- A LED light box does not guarantee the reproducibility of the pictures. The authors need to assess and validate the reproducibility of the LED (which one? White one? Power?) and also the camera device. They use a normal phone. Nevertheless, what happens with other cameras? This could be easily solved by adding standard color patch in the stripe of sensors. Then, the color changes can be calculated respect to the color patch. That color patch will calibrate any camera to the desired standards in any situation (different camera, different light intensity, different focal distance, etc) with the premise that the colors of the patch will no change or, at least, they are perfectly known. Also, check if with HSV or Lab colorspaces the results improve. And validate it with a strong and reliable validation.
- The comparison with 3-way PCA is totally useless. I agree that the data can be arranged in a 3-way manner. Nevertheless, this is not a trilinear problem. This is a bilinear problem. Therefore, it makes no sense to compare both methodologies (indeed, 3-way PCA includes a step of unfolding, converting it into a PCA). By the way, how do the authors calculate the Hotelling ellipsoids? They do not really look like real Hotelling boundaries of classes.
- Instead of 3-way PCA, the authors should include the results of a classification model (SIMCA, PLS-DA, Random Forest, or any other) to really assess the prediction capability of their methodology. PCA does not classify by itself. It needs class-modelling methods (e.g. SIMCA) in order to be a complete classification method. Therefore, use a real classification method.
Author Response
Dear Reviewer one,
Because of the articulate revision of this reviwer, we answered below each comments
This manuscript is, indeed, quite interesting. And I can see the huge applicability of a technically simple methodology. I like the scientific idea. Nevertheless, the manuscript, as it is, looks like a mere laboratory session for students. And do not get me wrong. I will clarify this statement. The authors need to create a solid methodology and validate it in order to create a successful methodology that can be used outside the laboratory environment. Only in that case this manuscript will make a real difference. For instance:
We thank the reviewer for the stimulating observations very much. Most of his/her suggestions have already planned in the project originated from this first research.
In a certain sense, it is true that it is a “laboratory session”. We do believe to have something suitable for an application. However, before transforming it into a device “that can be used outside the laboratory environment” we need to demonstrate that the background is consistent, the bases are robust and reliable. That is the reason why we kept so many sensors (six), being sure that in a final stage they will be reduced their number, and why we explored some of the most common meats. Our idea was to assess that the device can be extended and, in case, how to other foods, to understand how the different spoilage can affect the colours evolution.
In this sense, we did a lot of experimental work, and we organized this paper.
- Reproducibility of Dylon Colour Catcher formulation: It is too vague to use a commercial product and not validate its reproducibility. Either manufacture one by yourself following a specific formulation or verify the reproducibility of that commercial product by measuring different lots, different days the same lot, etc.
The reproducibility of different Dylon Colour Catcher was discussed in another paper of our ref [24,25] where we still applied CC, but for different purposes. For this stage of our research, we needed something cheap, practical, easy to prepare, to test the amount of ligand, the correct pH, to test if it works or not in the headspace in the presence of the meat, on which ratio and so on.
We described all preliminary experiments in old reference 8 (now 12). We are presently developing a synthesis of indicators linked to a polymeric substrate that is promising, but requires a chemical synthesis and an extrusion procedure that makes sense on the basis of the results presented here. We believed to have understood the real evolution of the products in the headspace, never described clearly because of a big misunderstanding about the role of BA’s that are present in the meat, but definitively overestimated in the headspace.
Since CC is definitively not our last solid support, the reproducibility is suitable for this stage of the research. For sure we will go on, we are already going on, saving when possible chemicals and waste of foods. In this sense, CC is an optimal compromise.
In the revised text we stressed more this aspect that maybe was not sufficiently augmented.
- A LED light box does not guarantee the reproducibility of the pictures. The authors need to assess and validate the reproducibility of the LED (which one? White one? Power?) and also the camera device. They use a normal phone. Nevertheless, what happens with other cameras? This could be easily solved by adding standard color patch in the stripe of sensors. Then, the color changes can be calculated respect to the color patch. That color patch will calibrate any camera to the desired standards in any situation (different camera, different light intensity, different focal distance, etc) with the premise that the colors of the patch will no change or, at least, they are perfectly known. Also, check if with HSV or Lab colorspaces the results improve. And validate it with a strong and reliable validation.
Reviewer is right, in the present version we add details about the LED performances, the reproducibility of the photos taken by the mobile phone was judged enough for the scope. In any case, the photos and the RGB values of 5 different phots of the same device is reported in the attached figure.
They are mean values of RGB data of five different photos of the same array, taken at different times during the day so that the illumination of the room is definitively different. The standard deviation is around 3.2% that for this purposes are more than enough.
Our goal is not to develop an app based of photos. Because in case you develop an app on mobile, it would to be of paramount importance to calibrate correctly the colour space, since it must work in the same way with any camera of any phone.
Here, our mind was to simply evaluate if the colour changes make sense with what is going on in the headspace, and our final goal is to find something suitable for a final naked eye detection. With this in our mind, does it make sense to use more complication space colour if with the simplest one we obtained the information we were looking for?
The reviewer is right to underline the problems that could arise, but referred to our aim, the quality of LED camera and RBG acquisition was judged suitable for the purposes, where the variability is definitively higher in the real samples that were used to develop the device.
- The comparison with 3-way PCA is totally useless. I agree that the data can be arranged in a 3-way manner. Nevertheless, this is not a trilinear problem. This is a bilinear problem. Therefore, it makes no sense to compare both methodologies (indeed, 3-way PCA includes a step of unfolding, converting it into a PCA). By the way, how do the authors calculate the Hotelling ellipsoids? They do not really look like real Hotelling boundaries of classes.
According to this statement, no 3 way-PCA makes sense. The only thing that we want to demonstrate, applying the 3-way PCA, and which is not at all evident by the single PCA of each kind of meat, is that the evolution of the substrate is similar and has sense. Considering that the data set was limited to the faster degradation, and consequently the comparison limited to the timeline of fish, the result is our opinion is not entirely odd, and demonstrate once more that the approach makes sense, that is consistent.
Nevertheless, thank to the reviewer’s observation, we understood that our explanation was not enough clear, and we added more detailed discussion in chapter 3.4, that, in our intent, justifies the employment of the 3 way-PCA. We hope to convince him/her…
The reviewer’s observation about the ellipsoids was right. It was our fault to do not adequately underline that they are not hotelling boundaries, they are simple ellipsoids, add to clarify the different step of the degradation. We apologize for that, now a comment is added. Much more essential, axes scales were added, to underline that are the same for PCA1 and PCA2, as done by the software.
- Instead of 3-way PCA, the authors should include the results of a classification model (SIMCA, PLS-DA, Random Forest, or any other) to really assess the prediction capability of their methodology. PCA does not classify by itself. It needs class-modelling methods (e.g. SIMCA) in order to be a complete classification method. Therefore, use a real classification method
As the reviewer knows, modelling and classification require several samples. At this phase of our research where we still have all the six indicators, and since for each of them three RBG triplets are required, we need at least 18 samples for each kind of meat, plus those required for validation (at least about 1/3 of those employed for training).
For sure we will apply a classification method, maybe LDA, or QDA, more than SIMCA, maybe more suitable for modelling, but in our mind we planned to do this after the selection of the definitive array of sensors, with indicators not anymore fixed on the Color Catcher, with the choice of indicators that will never be more than three. This part is currently going on and we were processing the very preliminary results just before the lockdown.

Reviewer 2 Report
The subject discussed in the article is interesting, but paper needs revision.
Specific comments:
- Manuscript should be prepared according to “Guide for Authors” i.e. line 189 – “150g” should be: “150 g “; “Table 1.- Experimental conditions...” should be “Table 1. Experimental conditions...” and next “Figure 2.-, Table 2.-“....; “.33.3 Sensitivity” should be “3.3 Sensitivity” etc.
- In the Materials and methods section, 1. Materials and Chemicals subsection should be separated.
- In my opinion chemical formulas of the dyes employed as sensing moieties (Figure 1) are unnecessary in this publication.
- The Authors should familiarize themselves with the proper format for References and make appropriate corrections.
Author Response
We want to thank this reviewer very much, we apologize for the improper format of many items, that now have been corrected.
All format of figures and tables, and the format of bibliography have been modified. The title of the chapter Materials and Chemicalswas added.
As suggested, we also took away figure 1.
Reviewer 3 Report
The manuscript submitted for review demonstrates a fast, relatively cheap, naked-eye technique for determining the spoilage rate of various types of meat. In a world of widespread consumerism and large-format stores focused on maximum profit and minimum loss, the ability to quickly confirm the freshness of food products, especially those with short shelf-life, is extremely important.
The overall quality of the presented manuscript is very good, however, few minor issues raise my concern.
How were the pigment concentrations shown in Table 1 determined? On what basis did the authors select such concentrations? I tried to find Reference #8, unfortunately, it is not available online yet, hence my question.
Apart from the Colour Catcher® mentioned in the text, did authors perform any tests carried out on similar products but from other manufacturers? I am curious whether the composition of this particular product is specific or whether other products of this type could also serve as a base for indicators.
Why in Table 5 there is a codfish sample missing?
Shouldn't the meat samples be of similar dimensions? Especially when attempts are made to determine the sensitivity of the proposed indicators. It seems to me that the larger the surface area of the examined sample, the faster changes will occur on it, thus a greater amount of compounds secreted from their surface, which may influence e.g. the rate of observed colorimetric changes on the indicators. A similar surface area of samples, at least in my opinion, would guarantee more reliable results in time.
Line 54: VOC abbreviation should be defined.
Line 302: Number of the paragraph is incorrect.
The presented research is extremely interesting and gives a lot of opportunities for implementation on an industrial scale. I hope that the authors manage to turn the presented ideas into real industrial solutions.
I recommend the manuscript a few minor corrections.
Author Response
We want to express our warm thanks for this revision. We hope to have answered in a satisfactory way, also in this case our answer is below each comment
How were the pigment concentrations shown in Table 1 determined? On what basis did the authors select such concentrations? I tried to find Reference #8, unfortunately, it is not available online yet, hence my question.
The concentrations are made from dilutions of stock solutions, built up from dissolution of the solid indicators, always chosen at highest purity. The idea for choosing the concentration for each indicator is to have the lowest concentration to increase the sensibility of the released VOCs, but anyway high enough to be clearly appreciated naked eye, especially when yellowish colours are involved. In any case for all, concentrations are well below saturation of positive charges of the CC.
We add a version of reference 9, now 12 for him/her.
Apart from the Colour Catcher® mentioned in the text, did authors perform any tests carried out on similar products but from other manufacturers? I am curious whether the composition of this particular product is specific or whether other products of this type could also serve as a base for indicators.
This selection was done at the early beginning. We also tested other products, previous conventional ones, as solid ion exchange resins and also membranes, easy to handle but definitively expensive. We moved to this laundry product by chance, and we found that the fabric of the classical Colour Catcher was definitively better, if compared to the other products (for instance OminoBianco Acchiappacolore) or to other following version of the same CC.
Why in Table 5 there is a codfish sample missing?
Yes, it is missing, and in the final development, it must be scheduled.
Actually, the idea of testing sensibility arose in case of meats, as a consequence, in preliminary stages, of odd results of samples with a lower mass. We observed that, in our reference supermarket, the tray for meats was bigger than that for fish, while the mass was similar. This means that meats VOCs are released in a bigger volume.
Moreover, it must be observed that the amount of VOCs varies with the type of meat and is always higher in fish samples. For these reasons here the sensitivity test was limited to meats and will be implemented in the early future.
Shouldn't the meat samples be of similar dimensions? Especially when attempts are made to determine the sensitivity of the proposed indicators. It seems to me that the larger the surface area of the examined sample, the faster changes will occur on it, thus a greater amount of compounds secreted from their surface, which may influence e.g. the rate of observed colorimetric changes on the indicators. A similar surface area of samples, at least in my opinion, would guarantee more reliable results in time.
Since we aimed to demonstrate the applicability of our device under the common retail conditions, we tested it on the most common selling trays. We discovered to contain almost a standard amount for each meat, possible for question related to the price. We agree that the amounts are significantly different. However, at this stage, since the device works, it was decided not to deepen this aspect, surely of interest and that will deserve further attention when we will move to the final label.
Line 54: VOC abbreviation should be defined.
The reviewer is right, and now abbreviation for VOCs was added.
Line 302: Number of the paragraph is incorrect.
Yes, now it has been changed.

Round 2
Reviewer 1 Report
After carefully analyzing all the replies to my queries, I must say that the authors have developed a dedicated revission. Still, I would have liked that the authors performed some of my suggested analytical changes. Nevertheless, they circumvented by saying to do it in next works related to this subjet. In any case, I believe that this manuscript is suitable for publication. I still like the scientific idea even though I would have loved to see in this manuscript more "action" instead a proof of concept.
Well, I hope they do it with a validated classification model in the near future.
There is still something that I do not like per-se. That is the comparison with 3-way PCA. Again, it is redundant and it does not offer much to the manuscript. But this could also be my personal appretiation. If the authors do not want to move from explorative methods, I would have rather compared PCA with any clustering method (or similar).
In any case, hoping that the authors keep working on this methodology, I think that the manuscript could be accepted for publication.
Author Response
We warmly thank this anonymous reviewer, also for his/her reply.
I hope to give him/her the satisfaction to revise, in the future, the development of this paper. For the purpose and the stage of this investigation, still explorative for the potential application, it was not the case to employ the number of samples needed to apply a classification method correctly.
If, as we hope, we develop a prototype of the intelligent label we are working for, all the experimental setup will be done as recommended by literature. Here, unsupervised explorative methods have been employed to the best of our knowledge. Clustering, even if popular, is depending too much on the choice of the distance and the similarity.